# Event-Based Angular Speed Measurement and Movement Monitoring

**DOI:** 10.3390/s22207963

**Published:** 2022-10-19

**Authors:** George Oliveira de Araújo Azevedo, Bruno José Torres Fernandes, Leandro Honorato de Souza Silva, Agostinho Freire, Rogério Pontes de Araújo, Francisco Cruz

**Affiliations:** 1Escola Politécnica de Pernambuco, University of Pernambuco, Recife 50720-001, Brazil; 2Unidade Acadêmica da Área de Indústria, Federal Institute of Paraíba, Cajazeiras 58900-000, Brazil; 3School of Computer Science and Engineering, University of New South Wales, Sydney 1466, Australia; 4Escuela de Ingeniería, Universidad Central de Chile, Santiago 8330601, Chile

**Keywords:** event-based vision, angular speed, rotational measurement

## Abstract

Computer vision techniques can monitor the rotational speed of rotating equipment or machines to understand their working conditions and prevent failures. Such techniques are highly precise, contactless, and potentially suitable for applications without massive setup changes. However, traditional vision sensors collect a significant amount of data to process and measure the rotation of high-speed systems, and they are susceptible to motion blur. This work proposes a new method for measuring rotational speed processing event-based data applied to high-speed systems using a neuromorphic sensor. This sensor produces event-based data and is designed to work with high temporal resolution and high dynamic range. The main advantages of the Event-based Angular Speed Measurement (EB-ASM) method are the high dynamic range, the absence of motion blurring, and the possibility of measuring multiple rotations simultaneously with a single device. The proposed method uses the time difference between spikes in a Kernel or Window selected in the sensor frame range. It is evaluated in two experimental scenarios by measuring a fan rotational speed and a Router Computer Numerical Control (CNC) spindle. The results compare measurements with a calibrated digital photo-tachometer. Based on the performed tests, the EB-ASM can measure the rotational speed with a mean absolute error of less than 0.2% for both scenarios.

## 1. Introduction

Rotating machines are present in many industry areas, like machining, turbines, motors, gears, and shafts, and the rotation speed must be under control to prevent unexpected mechanical failures [1]. Measurement system for instantaneous rotational speed (IRS) can characterize faults in mechanical systems [2], such as gears [3,4,5], engines [6,7,8], shaft torsional vibrations [9], bearings [10,11,12], high rotating speed rotating propellers [1], estimate life cycle of machines, and evaluate their work conditions. Therefore, a reliable measurement system is required to control these parts’ life cycles.

In the rotational speed measurement systems area, there are two categories, the contact ones and non-contact systems [13]. The contact measurement system requires to be mounted on the rotating object. It often causes wear on the system over time, and there is an influence on rotating parts due to the additional mass [14,15]. There are many types of non-contact measurement systems, such as optical encoders [16], electrical [17], magnetic induction [18,19], and vibration signal analysis [20]. However, these sensors present limitations and need adaptations for each industrial environment application [15]. For example, electrostatic signals’ measurement depends on the rotor material and its roughness. At the same time, vibration techniques perform a high-accuracy measurement. However, its performance degrades due to harmonic torque components [18] even in the harsh industrial environment due to the great number of vibration noise sources. Another way to measure rotational speed without contact is the use of computer vision.

Vision-based measurement systems have their application range increased widely in the last years due to advances in software and hardware technologies [21]. They are used for controlling applications such as robotics [22,23] and automation [24], which requires a precision level to guarantee the system stability and control. The performance of high-speed control systems depends on the accuracy and the latency perception to achieve the desired agility [25]. Due to advances in imaging sensors and image processing algorithms, the use of vision-based sensors can change the paradigms of monitoring machines. By visual analysis, it is possible to extract information about the environment and use it as a multi-sensor tracking multiple regions simultaneously [26]. For high rotational speed, the measurement system requires a high-speed camera such as the one used by [27], and it is also necessary to track a marker in the rotating surface. However, these techniques are still susceptible to motion blur since the camera frame rate should be superior to the rotational frequency. That might not be possible to measure on some machines, or it might not be possible to include a specific marker for tracking.

In contrast to the traditional frame-based vision systems, the event-based cameras use neuromorphic sensors to record illuminations changes with microsecond accuracy per pixel, while traditional camera record a fixed number of pixels per frame at a constant frame rate [28]. The output data from neuromorphic sensors are stored as a stream of events, and novel processing methods are required [29]. This new sensor has promising applications for high-speed systems because of its fast response. Its inspiration is the biological vision to overcome traditional camera limitations. The main advantages are the fast response, high dynamic range, low power consumption, and few stored data [29,30,31,32]. The Dynamic Vision Sensor (DVS) output is a sequence of asynchronous events rather than frames [33]. The DVS transmits independent brightness changes for each pixel called events. Compared to the traditional frame-based camera output, the DVS or Neuromorphic Vision Sensors (NVS) produce a set of events or spikes at pixel coordinates, presented by the brightness changes, positive or negative, and their time occurrence. Neuromorphic sensors are used for purposes such as pose tracking for high-speed maneuvers like quadrotors [25], event camera angular velocity estimation [30], reconstruction of image intensity images from high speed and high dynamic range video [34], odometry system based on events to estimate displacement [33]. Other applications, including object recognition, depth estimation, simultaneous localization, and mapping, also use this kind of sensor [29]. Currently, the major obstacle to using advanced machine learning algorithms for classification and recognition tasks with event-based systems is the absence of available streams datasets [28,32].

This work proposes a new method to measure rotational speed using data from an event-based sensor. It is called Event-Based Angular Speed Measurement (EB-ASM), which uses an event-based vision sensor stream to measure the rotational speed based on the elapsed time between positive and negative events. To the best of our knowledge, no previous research proposes using dynamic vision sensors (DVS) to measure the angular velocity of rotational objects. Thus, this paper shows the proposed method evaluation in an experimental environment by measuring a multi-blade propeller rotation and a router CNC spindle rotation. In addition, the EB-ASM results in both experimental environments are compared with the rotation measurement from a digital tachometer. A detailed explanation of the data processing and algorithm stages is developed in this text. Furthermore, the available datasets can evaluate other rotational measurement methods with event-based data proposed in the future.

This paper is organized as follows: Section 2 outlines the measurement principles for rotational speed measurement, explains the sensor data generation, and how the proposed algorithm works. Section 3 describes the experiments to evaluate the measurement system. Section 4 presents the measurement results, and the Section 5 discuss the results.

## 2. Event-based Angular Speed Measurement (EB-ASM)

The general schematic diagram of the measurement system is shown in Figure 1, where any rotating device or machine can replace the rotating fan blades. The reference measurement device in Figure 1 is a calibrated measuring instrument, and its output is compared to the proposed algorithm measurements. The system consists of any rotating object with at least one visible moving edge, such as a plane propeller, a wheel, a gear, a machine spindle, and the event-based vision sensor to collect the stream data. This sensor detects brightness changes when the object is rotating. The proposed algorithm analyses a fixed region and estimates the object rotation.

### 2.1. Rotational Speed Measurement Principle

An Angular Speed measurement system depends on reliable aspects such as measurement principle, sensor selection, sensor signal conditioning, performance parameters, and analysis [2]. The angular speed is based on the elapsed time between pulses and the angular displacement, predefined by the number of symmetrical parts in the measuring device. The following equation shows the measurement principle in the proposed method
(1)$$\omega =\frac{\Delta \phi }{\Delta t}$$
where Δϕ is the angular displacement, and Δt is the time duration. If the system has multiple rotating patterns with equal angular displacement, the rotational speed is given by
(2)$$\Delta \phi =\frac{360^{\circ }}{N_{b}}$$
where $N_{b}$ is the number of rotating patterns, for example, the number of fan blades. Moreover, the elapsed time is the difference between $t_{i}$ and $t_{f}$. The sensor activation occurs by the pattern passing in the measurement region. For counting the elapsed time in μs, the rotational speed in rotations per minute (rpm) is calculated through the Equations (Equation 1) and (Equation 2) and shown as
(3)$$n=\frac{60}{N_{b}\left(t_{f}-t_{i}\right)\cdot 10^{6}}$$
where $t_{f}$ and $t_{i}$ are the final and initial time, respectively, in μs, for the pattern detection in the selected measurement region.

### 2.2. Event-Based Vision Sensor Description

The event-based sensor presents a different way to record visual information than standard camera sensors. Instead of capturing a full frame at a specific rate, the event camera’s pixels output is generated due to brightness change in continuous time registered and transmitted asynchronously. Each pixel activation, called a spike, transmits its coordinates, timestamp, and event polarity. The events are this set of information about a pixel activation with brightness changes ($I_{i,j}^{\prime }$) higher than the threshold. The sensor sensibility to brightness changes is a parameter called the threshold. The user defines it during its setup and means the lowest level of luminance changes that led to events. Events polarity ($I_{k}$) are related to the threshold by
(4)$$I_{k}=\left\{\begin{array}{cc} +1 & I_{i,j}^{\prime }>T_{h}\\ -1 & I_{i,j}^{\prime }<-T_{h} \end{array}\right.$$

The brightness changes source are moving surfaces or object edges, and the event camera captures this information as independent pixels output. The sensor output for each event *k* is a tuple $e_{k}=\left(t_{k},x_{k},y_{k},I_{k}\right)$, where $x_{k}$ and $y_{k}$ are the pixel spatial coordinates, $t_{k}$ is the temporal coordinate of the event, and $I_{k}$ is the binary event polarity. Clearly, instead of outputting brightness intensity for every pixel in the frame-based sensor, the event-based sensor output only triggered pixels by brightness changes.

Due to the sensor’s sensibility to brightness changes, its output depends on the amount and speed of moving edges to generate output data. These generated data are transmitted by an Address-event Representation (AER) and called event-driven as their output depends on the amount of motion detected by the camera. In addition, the time to sweep all activated pixels limits the maximum rotational speed to measure, called temporal resolution. The temporal resolution for an event-based camera is compared to the frame rate of a frame-based camera.

### 2.3. Event-Based Angular Speed Measurement Algorithm

A moving object in front of an event-based sensor generates a data stream with positive and negative spikes due to its periodically moving edges. Brightness increases result in positive spikes and a brightness decrease in negative spikes. The transition between these signals is proportional to the rotational speed. This information is used to measure the rotational speed in the proposed algorithm. In addition, this approach analyzes regions in the image during the stream to measure the elapsed time between opposite spikes in the selected area.

The main point of the method is select a tiny region in the frame and count the time between events in this area. Due to the event-driven behavior of the output, the amount of data grows as the number of moving surfaces and the moving speed increase. Furthermore, due to the redundant data processing, selecting a small region of interest in the frame is necessary to reduce data processing. This region is called Kernel (ω) with size ($S_{\omega }$), and it works like a fixed window in the frame sensor.

The rotational speed measurement process is divided into three steps. The first is the data collection, followed by the selection of events inside the fixed position and size Kernel. Then, the algorithm uses the events signal and timestamps to calculate angular displacement and the elapsed time between edges. The last step uses the elapsed time to measure the angular rotational speed.

Therefore, some essential points to discuss for events data processing, like the presence of noise events, nearby events with the same signal and different timestamps, and the angular displacement setup.

A data sequence contains the moving edges as spikes with signal ($I_{k}$) for each event and timestamps with tiny differences. We ignore the time difference and consider only the first event timestamp of the edge. The rotational speed calculus does not consider isolated events that present different signals. For this purpose, the algorithm considers a signal change when most events in the measurement region show an opposite spike signal. The first timestamp of the signal transition is registered for rotational speed calculus using Equation (Equation 3). Figure 2 shows three different flows of events occurring inside the Kernel, most events are positive in the first two situations, and most are negative in the third. From the second to the third box, there is a transition of spikes signal, and the first timestamp should be saved. The transition time is registered when most events inside the Kernel turn negative, and the timestamp of the first event is saved to calculate the rotational speed.

To summarize the method, Algorithm 1 shows the pseudo-code with the main steps. All events mentioned here are inside the kernel, the first spike signal is saved, and when the signal of most events changes and returns to the first one saved, there is the elapsed time for an angular displacement pattern. This approach does not require any information about the center of rotation coordinates. However, in the Kernel positioning, it is required to locate it in an image region where there are events with different signals over time.
**Algorithm 1** Rotational Speed Measurement Algorithm1:$e_{i}$, $e_{m}$, $e_{f}$←*⌀*2:**for**$e_{k}=\left(t_{k},x_{k},y_{k},I_{k}\right)$ in events stream **do**3:    **while** $e_{i}==⌀$**do**                    ▹ Loop to get the initial event4:        **if** $e_{k}$ is inside the Kernel **then**5:           $e_{i}\leftarrow e_{k}$6:        **end if**7:    **end while**8:    **while** $e_{m}==⌀$**do**                  ▹ Loop to get the middle event9:        **if** $I_{k}\neq I_{i}$ and $e_{k}$ is inside the Kernel **then**10:           $e_{m}\leftarrow e_{k}$11:        **end if**12:    **end while**13:    **while** $e_{f}==⌀$**do**                    ▹ Loop to get the final event14:        **if** $I_{k}\neq I_{m}$ and $e_{k}$ is inside the Kernel **then**15:           $e_{f}\leftarrow e_{k}$16:           Compute the elapsed time $\Delta t=t_{f}-t_{i}$17:           Compute rotational speed $n=60.10^{6}/\left(N_{b}\Delta t\right)$ at rpm18:           $e_{i}\leftarrow e_{m}$19:           $e_{m}\leftarrow e_{f}$20:           $e_{f}\leftarrow ⌀$21:        **end if**22:    **end while**23:**end for**

This approach updates the rotational speed for every set of spikes with the first registered signal. Only one Kernel is used for angular speed measurement, but it is easily changed to use more Kernels set to different objects and measure different rotational speeds. In the following sections, for the method validation, one rotation is used.

## 3. Experimental Protocol

The Dynamic Vision Sensor collects data on rotating objects, which are used to evaluate the EB-ASM method. Inivation manufactures the sensor used in this work is the DVXplorer Lite with 320 × 240 spatial resolution and 200 μs temporal resolution. Tests are performed in two different scenarios. In the first test, the sensor is positioned perpendicular to the rotation plane of a multiple-blade fan and connected. The record stream of data is shown in Figure 3, presenting the events in a Spatio-temporal image. The second scenario measures an angular speed of a Router CNC spindle, and the sensor is positioned beside the spindle shaft. This setup in Figure 4 shows the spindle rotation measurement, and it is evaluated for three different speeds.

The timestamps ($t_{k}$) are recorded in microsecond resolution (μs), and the spatial pixel coordinates are recorded accordingly to the pixel position in the spatial resolution. The $x_{k}$ coordinate is a value between 0 and 319 and $y_{k}$ coordinate between 0 and 239. The temporal resolution corresponds to the period to read out all pixels. Thus, the minimum time difference measured for the rotation system is 200 μs. Figure 3 clearly shows the temporal resolution of this sensor by the sloping clouds of data, where red dots represent positive spikes and purple dots represents negative spikes. Figure 5 represents the top view of Figure 3 and permits see the spikes projected in a single frame.

The EB-ASM method measures the rotational speed with the event-based sensor and uses a timer/counter-based technique based on measuring an elapsed time between successive pulses [2]. Furthermore, as we use a dynamic vision sensor (DVS), any region of the activated pixels can be analyzed. However, it is necessary to select a region where there are events. Thus, the best region is where there is an object moving. The great advantage of this method is processing a small portion of the sensor data. Thus, a reduced processing time is achieved. Positive and negative events occur when passing each rotating pattern due to brightness changes in the analyzed region. However, noisy events at each passage can confuse the measurement when analyzing a single point. The Kernel strategy helps to avoid the influence of noisy events caused by natural luminosity. The trend of events inside the Kernel determines the majority spike’s signal.

The recorded data of a fan show events with the same timestamp or timestamps are significantly closer and smaller than the temporal resolution, meaning they are almost simultaneous events.

As shown in Figure 3, simultaneous events have the same timestamp or timestamps significantly closer and smaller than the temporal resolution. Thus, we use the elapsed time between successive pulses to measure the rotational speed and the pixels inside the Kernel to guarantee if they are positive or negative. The first timestamp of an event in the selected region starts the timer count and stops when the signal changes twice.

Section 2 presents two steps to set up the measurement system. The first one is to set the sensor parameters, such as the luminosity threshold, which influences noisy events in the recorded data. This parameter is not analyzed in this paper because it is possible to record moving edges for rotational systems, even with the wrong threshold selection during sensor setup. The second step is to analyze the recorded data processing to get the angular speed of the system, and it requires two parameters, the Kernel position, and size. Then each event in these pixels region is recorded in a list to verify its spike value and timestamp.

The proposed method is compared to a reference speed measured with a digital photo-tachometer model UT372 from UNI-T, with reliability and stability to measure the rotational speed of points in machines. It ranges from 10 to 99,999 rotations per minute with a 0.01 rpm resolution and ±0.4% + 2 digits accuracy. The sampling rate is around 6 data per second. Moreover, its distance from the rotational machine should be between 50 and 200 millimeters. The photo-tachometer works with a light source pointed to a reflective sticker placed in the rotating element. The tachometer sensor is triggered as the light is reflected by the light source. The rotational speed measurement is computed to measure the rate of this signal. However, the digital tachometer requires a limited distance positioning from the rotating machine with a reflective surface. Therefore, while the EB-ASM can measure a broader range of rotational speed, it is possible to use visual information to collect data about the system’s movement.

The rotating fan has eight blades, a maximum 150 W power, and a three-speed selector. However, the fan speeds are not controlled and can fluctuate depending on factors such as air resistance or variations in electrical current. Therefore, the digital photo-tachometer measurement registers these speed variations. Then, the event-based measurement system is compared with the tachometer speeds to verify its measurement error.

The EB-ASM is also evaluated in a Router CNC machine with constant and controllable speed, and the spindle rotation without a machining tool is measured. Tests are performed for three constant speeds, and the photo-tachometer uses a reflective tag added to the spindle. The event-based sensor measures the rotating system from a side view of the rotating patterns in the spindle end.

As the EB-ASM method calculates the rotation by the angular displacement of each rotating pattern individually, it is subject to high-frequency noises between the specific ones. So, to reduce this type of noise, a low-pass filter was added to the measurement system.

Experimental results are shown in the next section to verify this method’s availability to measure rotational speed with the event-based system as a non-contact sensor.

## 4. Results

The EB-ASM approach is validated by measuring the Mean Absolute Error (MAE). The rotational speed processed from the even-based vision output data is obtained from the elapsed time between spikes with different signals. For example, one experimental setup uses fan blades as a rotating pattern at different speeds. The other one measures the rotation of a CNC spindle at different speeds.

The measurement system setup requires selecting three parameters: The Kernel position, size, and the number of rotating patterns. The Kernel position should be selected in a region with significant brightness changes during the movement of the blades, as shown in Figure 5 where the *x*,*y* position is 190 and 175. One could select any position at the sensor field of view where there are positive and negative event patterns changes over time. Its size influence is shown in Table 1 where the Kernel size is presented with the mean absolute error for the rotating fan setup. The Kernel size 8 × 8 is the maximum admitted without the presence of highly noisy signals. The size 10 × 10 presents a higher standard deviation and noisy measurements due to the higher number of disturbance events in the Kernel it is subjected to, shown in Figure 6. With the Kernel 8 × 8 there are 64 analyzed pixels, and with the window size 10 × 10 there are 100 pixels. This size difference is sufficient to increase the MAE almost seven times comparing these two biggest tested windows.

The sensor data generation depends on the dynamics of the rotating objects and their rotational speed because it represents a high number of brightness changes on the sensor’s pixels. Table 2 shows the time recorded at each speed of the fan, the number of events generated, and the number of events per second. The number of rotating parts significantly influences the amount of data. In addition, the brightness setting on the blades and the relative position between the sensor and the rotating parts influence the number of activated pixels during movement. However, the brightness influence is not evaluated in this work. The main goal is to present the measurement system with this new sensor and the algorithm.

The EB-ASM method measures the rotation of a fan with eight symmetrical blades. The selected parameters are the Kernel position and size. The size influences are shown in Figure 6 and Table 1. The number of symmetrical parts is required for the measurement system setup. Otherwise, the system measures multiple rotations.

The method evaluation compares the EB-ASM output values with the digital tachometer measurement. The Mean Absolute Error (MAE) for each rotational speed tested in this work is shown in Table 3. Figure 7 shows the measurement values compared with the reference rotational speed of the fan blades using the EB-ASM method. The fan has three different stages of rotating speed, called V1, V2, and V3. Although all the results presented variable rotations due to dragging on the blades, electrical current fluctuations influence the fan motor. Therefore, the results present the validation of the EB-ASM method and its accuracy for the recorded data, even with variable rotational speed.

Another experimental setup is the measurement of the router CNC spindle to validate the EB-ASM for a rotating machine as a future application for machining processes. In this case, there is no influence of the environment on the spindle rotation, then the rotational speed does not fluctuate during the experiments, as shown in Figure 8, Figure 9 and Figure 10. The mean error in the measurements is about 0.15%, in Table 4, for the fastest rotation with results in Figure 10, but the high frequency noise increase with the rotation. It is important to mention that the event-based sensor is positioned in a side view of the spindle and shows versatility in the sensor positioning compared with the fan setup.

## 5. Conclusions

In this work, a method for rotational speed measurement called EB-ASM is proposed and evaluated in two experimental scenarios. An experimental test with a fan and another with a Router CNC spindle are used for the method evaluation.

It is applicable for rotating systems with edges such as fans, gears, spindles, and other dynamic systems with low and high speeds. As far as we know, this is the first proposal to use event-based sensors to measure the rotational speed of machines. The main advantages of this measurement system are the possibility to measure far from the rotating object, it does not require hardware implementations in the machine, and the vision information of the sensor can act as a safety system, stopping moving parts from an unaware person.

This rotational measurement approach has the advantage of measuring high-speed rotations without requiring processing a large amount of data, reduced time processing, and is not influenced by axis rotation movements. In addition, the sensor has a different type of data than traditional cameras and can measure rotational speed as a non-contact sensor.

The multiple rotating patterns give redundant data in experimental tests and increase the obtained information to process. Thus, to reduce the time processing, the proposed method requires the selection of a delimited region, called a Kernel, and this region size influences the noise of the results. The maximum Kernel size without noise measurement for the tested data was a square with 8 × 8 pixels.

The proposed method could achieve low error rates in both experimental environments, becoming a reliable solution for measuring high-speed rotating systems. Although, changes in lighting and selection of the analyzed region can significantly influence the amount of data to process. However, it does not influence the measurement performance. The measured error rates are lower than 0.2% for both cases.

For future work, the accuracy of this method should be tested with multiple rotating speeds and measured in real time for online monitoring. In addition, it is recommended to validate with other machines, such as wind power generator blades, other machining equipment, and rotating parts on industrial plants. Finally, the best proposition for this method is for applications for measuring multiple rotation objects simultaneously, which allows a significant advantage compared to traditional techniques for measuring rotation. Furthermore, the visual data allow for more information about the rotating machine, such as the integration with the supervisory system and the monitoring in the industry 4.0 context.

## Figures and Tables

**Figure 1 sensors-22-07963-f001:**
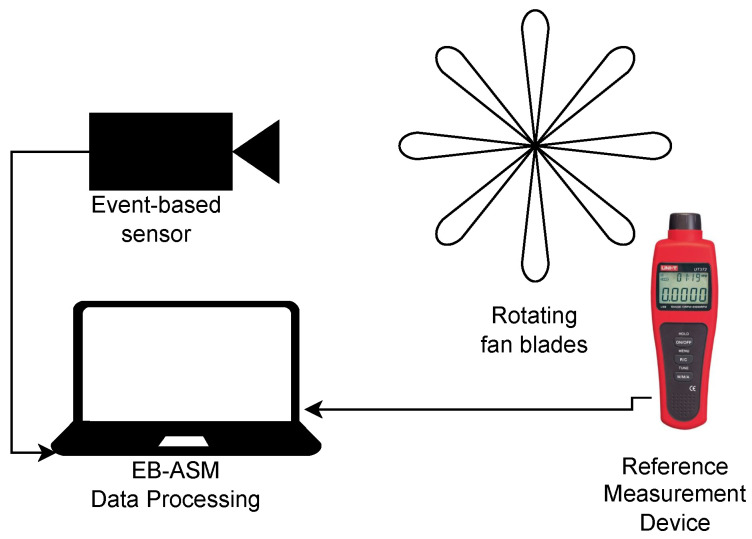
Schematic diagram of the proposed measurement system setup to record stream data.

**Figure 2 sensors-22-07963-f002:**
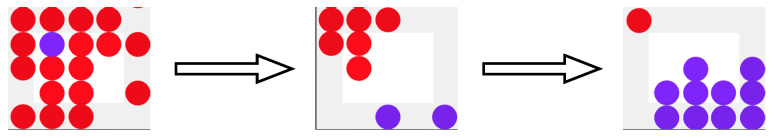
Flow of spikes inside the kernel during three different moments (positive spikes in red and negative spikes in purple).

**Figure 3 sensors-22-07963-f003:**
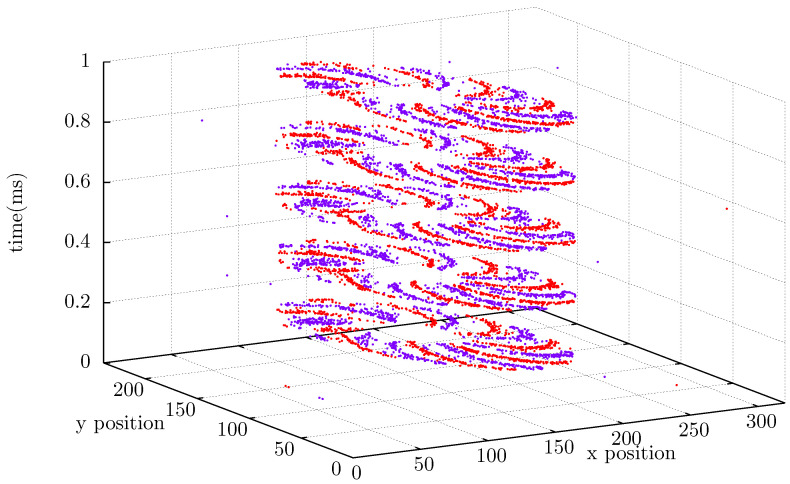
Stream data view of fan events with eight blades rotating. Coordinates *x* and *y* in the horizontal plane and timestamp in ms at the vertical axis.

**Figure 4 sensors-22-07963-f004:**
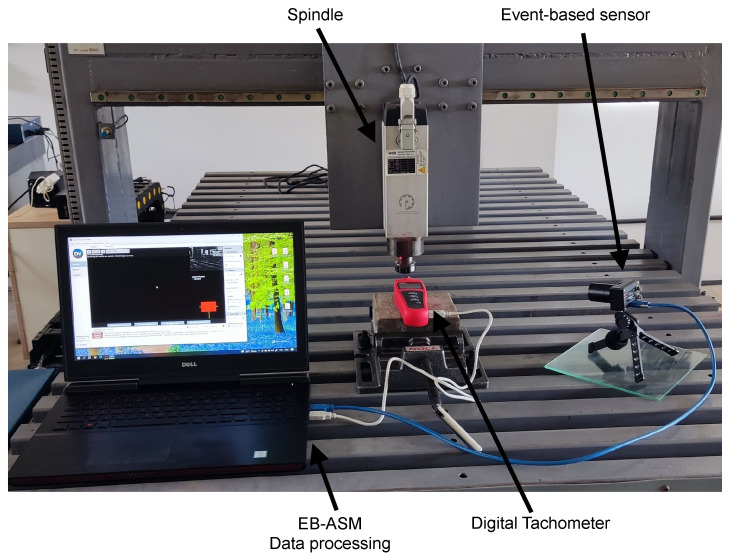
Experimental setup for measuring Router CNC spindle angular speed.

**Figure 5 sensors-22-07963-f005:**
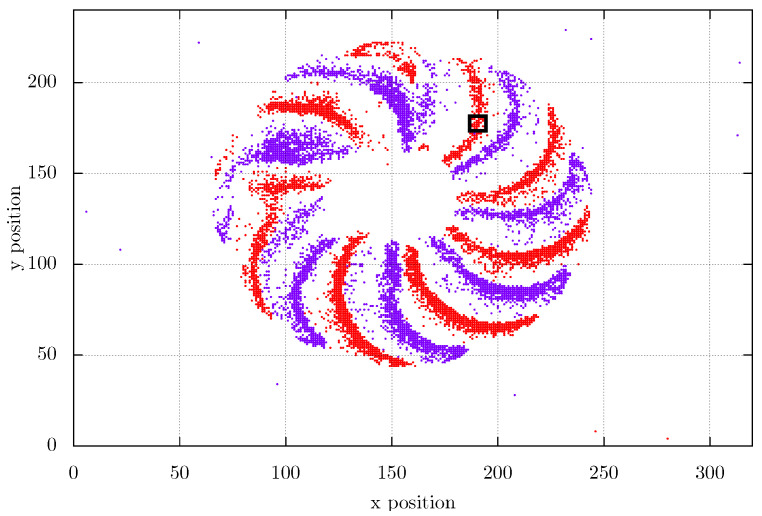
Stream data at 1 ms projected in the same plane and the Kernel is the small square.

**Figure 6 sensors-22-07963-f006:**
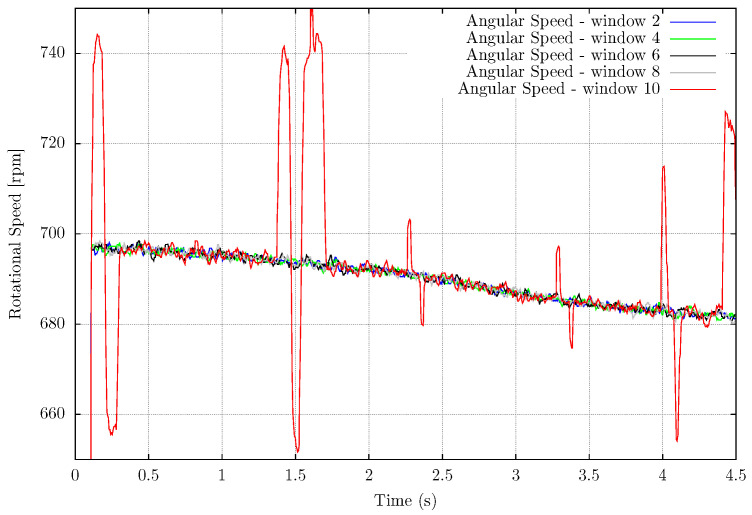
Event-based Angular Speed Measurements applied for a rotating fan varying the Kernel size.

**Figure 7 sensors-22-07963-f007:**
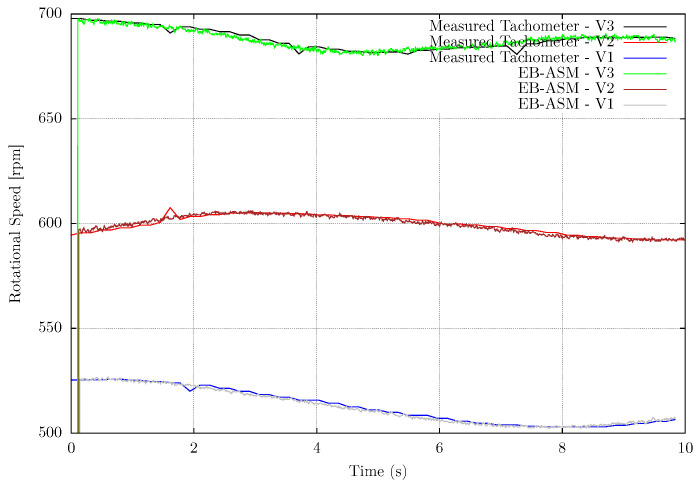
Event-based Angular Speed Measurements applied for a rotating fan at three different speed stages, named V1, V2 and V3.

**Figure 8 sensors-22-07963-f008:**
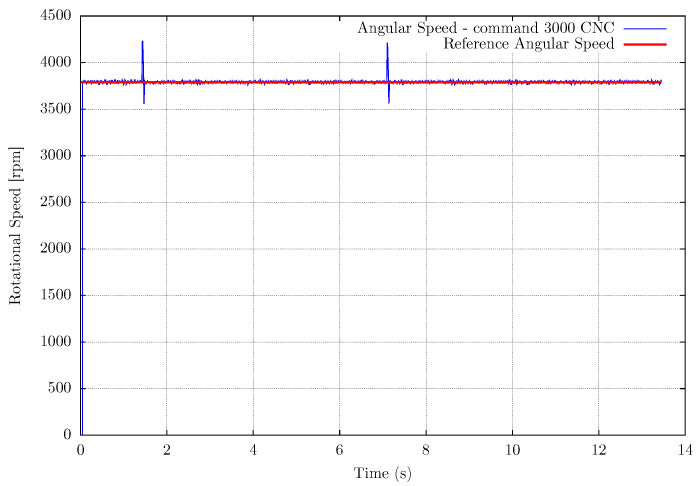
Measurement results for Router CNC spindle measurement at 3000 machining numerical command.

**Figure 9 sensors-22-07963-f009:**
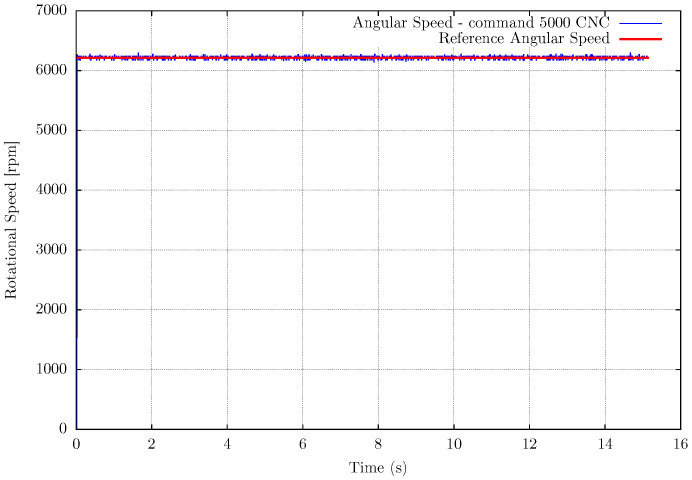
Measurement results for Router CNC spindle measurement at 5000 machining numerical command.

**Figure 10 sensors-22-07963-f010:**
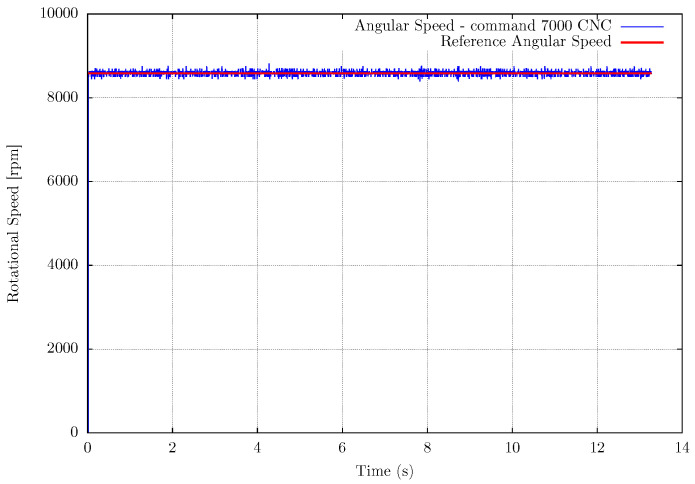
Measurement results for Router CNC spindle measurement at 7000 machining numerical command.

**Table 1 sensors-22-07963-t001:** Influence of measurement window size for reference speed rotation.

Window Size (Pixels)	Mean Absolute Error (MAE)	Standard Deviation
2 × 2	1.0733	1.4176
4 × 4	1.2215	1.5868
6 × 6	1.1296	1.4714
8 × 8	1.2145	1.5116
10 × 10	7.1990	15.4132

**Table 2 sensors-22-07963-t002:** Amount of data generated by the sensor at different rotational speeds.

Rotation Speed Stage (rpm)	Time (s)	Number of Events	Events per Second
Stage 1	5.569961	127,224,558	7,696,585.9
Stage 2	5.709986	138,440,305	8,604,121.4
Stage 3	4.719964	150,908,575	9,575,440.2

**Table 3 sensors-22-07963-t003:** Results for the MAE for the measurement method experimental evaluations with the fan.

Rotational Speed Stage (rpm)	Mean Absolute Error (MAE)	Standard Deviation
Stage 1	0.7419	0.8486
Stage 2	0.7857	1.0648
Stage 3	1.2215	1.5869

**Table 4 sensors-22-07963-t004:** Results for the MAE for the measurement method experimental evaluations with the Router CNC Spindle.

Rotation CNC Command	Mean Reference Rotational Speed (rpm)	Mean Measurement (MM)
3000	3790.5	3789.96
5000	6213.9	6207.8
7000	8582.4	8569.19

## Data Availability

The data presented in this study are available on request from the corresponding author.

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
