# Peer review of "Event-Based Angular Speed Measurement and Movement Monitoring"

_sensors, 2022, doi:10.3390/s22207963_

Round 1

Reviewer 1 Report

Article

EVENT-BASED ANGULAR SPEED MEASUREMENT AND MOVEMENT MONITORING

The paper presents an approach to the contactless measurement of rotational speed using an event-based camera. An algorithm is included to process the asynchronized event data on a selected kernel (subset/window). Experimental data with different scenarios have been conducted to demonstrate the method. The results are also validated by comparing the measurement with a different type of instrument.

There are some comments below.

Figure 4 image quality can be improved. It is a bit small on the paper.

How about the image coordinate for the centre of rotation? Would this location affect the results from a kernel?

Table 1, what may be the reasons that kernel size greater than 8 yields unreliable results? How can such size be determined in a more systematic way for any other application?

Line 284, there are unexpected question marks.

The rotation speeds’ determination appears to be offline. i.e. the events in figure 5 were captured first, and then the data were post-processed by a kernel algorithm to estimate the speed. If this is the case, how could a such method be made ‘online’ as well to achieve the purpose of ‘speed monitoring’ as claimed in the title? 

Reviewer 2 Report

In this work, the authors report an event-based angular speed measurement method. Due to the high temporal resolution and high dynamic range of the event camera, the rotating object does not generate motion blurring within a certain speed range. The event-based sensor collects the rotational motion signal of the object, and the rotation speed of the object can be calculated according to the response time. The mean absolute error of the measured rotational speed is less than 0.2%. The topic is interesting and the reported method may have potential applications in practical mechanical systems. Before I recommend its publication in Sensors, the authors should clarify the following technical questions.

1. In lines 117-118, is the brightness threshold related to the rotational speed of object and the dynamic range of event camera? How does the brightness threshold decide the theoretical upper limit of the measurable rotational speed?

2. in lines 126-127, the output depends on the amount and speed of moving edges. In my understanding, the measured results should be related to the radial position. The authors may explain how to choose a proper radial position.

3. In Table 1, the Kernel size of 10 presents a significantly larger error, which is attributed to the disturbance events as shown in Fig. 6. However, it is still confusing where these disturbance events come from abruptly. In comparison to Kernel size of 8, only two more pixels are used but the standard deviation increases from 1.5 to 15. It is better to analyze the possible noise sources.

4. In lines 253-254, “The Kernel position should be selected in a region with significant brightness changes during the movement of the blades”. How to define significant brightness changes in experiment?

5. Is the proposed method capable of measuring a variable rotational speed?

6. Some typos should be corrected. For example, in line 239, AB-ASM may be corrected to EB-ASM.

Reviewer 3 Report

modify introduction 

Reviewer 4 Report

1. English language should be improved. 

2. Accuracy of this method should be tested, it is explained theoretically. 

3. CPU fan speed has measured, gears, and spindles case study should be included. 

4. Working principle of proposed sensors should be explained using existing literature. 

5. Comparative analysis between proposed method and existing method using a case study. Compare the errors of the system. 
